# An Optofluidic Young Interferometer for Electrokinetic Transport-Coupled Biosensing

**DOI:** 10.3390/mi15070861

**Published:** 2024-06-30

**Authors:** Elisabetta Labella, Ruchi Gupta

**Affiliations:** School of Chemistry, University of Birmingham, Birmingham B15 2TT, UK

**Keywords:** Young interferometer, electrokinetic, microfluidics, biosensing, streptavidin

## Abstract

Label-free optical biosensors, such as interferometers, can provide a comparable limit of detection to widely used enzyme-linked immunosorbent assays while minimizing the number of steps and reducing false positives/negatives. In 2020, the authors reported on a novel optofluidic Young interferometer (YI) that could provide real-time spatial information on refractive index changes occurring along the length of the sensor and reference channels. Herein, we exploit these features of the YI to study interactions of biomolecules with recognition elements immobilized in selected regions of agarose gel in the sensor channel. We show that the YI is well suited for the biosensing of an exemplar biomolecule, streptavidin, in the absence and presence of the bovine serum albumin interferent. Equally, we couple the YI with electrokinetic transport to reduce the time needed for biosensing.

## 1. Introduction

Historically, the detection of biomolecules has been achieved using labeled techniques with an enzyme-linked immunosorbent assay (ELISA) as the gold-standard method offering a limit of detection of 1 pM [1]. ELISA, however, is time-consuming because of multiple adding and washing steps and the use of labels, which increases the probability of false positives/negatives. Over the last two decades, label-free biosensors have emerged that can not only overcome the limitations of labeled techniques but, in some cases, offer the same limit of detection as their labeled predecessors. In contrast to their predecessors, label-free biosensors detect biomolecules based on a universal property, such as the refractive index (RI) [2,3] and mass [4]. 

Label-free optical biosensors rely on changes in RI for the detection of biomolecules. Examples of label-free optical biosensors include those based on surface plasmon resonance [5], leaky waveguides [6,7,8], spectroscopic methods [9,10], and interferometry [11,12,13,14]. Among the label-free optical biosensors, interferometers offer the lowest limit of detection. A common type of interferometer is the Young interferometer (YI), where incoming light is split into two beams. One of the light beams interacts with a sample solution and the other with a reference solution. The two beams then interact with each other to produce a periodic pattern of intensity called interference fringes. The position of these interference fringes shifts as the RI of the sample solution changes. RI is proportional to the concentration of sample solutions, and hence, an analyte concentration can be determined by measuring shifts in interference fringes. 

The majority of the YIs measure either an RI change at a single point or an average change in RI over a specific distance. In contrast, in 2020, we reported a novel YI for the real-time imaging of RI changes along the length of microfluidic channels [14]. Furthermore, the majority of the YIs are based on integrated optics where the two light beams travel in waveguides, and only the evanescent field interacts with sample/reference solutions [15,16,17]. In contrast, the YI reported by the authors was optofluidic, where light beams interact with the entire depth of solutions in the sample and reference channels of a microfluidic device [13]. 

There has been significant growth in microfluidic devices, including those based on electrokinetic forces, for the rapid transport and separation of electrically charged biomolecules, proteins, and DNA [18,19]. The detection of biomolecules in microfluidic electrokinetic devices is typically performed using absorbance, fluorescence, and mass spectrometry. There are so far only two reports on the integration of electrokinetic transport with interferometry detection. One of these was by the authors, where the optofluidic YI was coupled with electrokinetic transport to study the migration of a model protein under an applied electric field [14]. The other report was by Yang et al. [20], who monitored the electrokinetic separation of proteins using a Mach-Zehnder interferometer (MZI). 

Our previous work [14] reported the basic principle of a novel optofluidic YI and showed that our YI can be used to study the electrokinetic transport of a protein. Herein, for the first time, we show that our optofluidic YI, integrated with electrokinetic transport, is suited for biosensing. We filled the sample and reference channels of our microfluidic device with a hydrogel with recognition elements immobilized in specific regions of the hydrogel in the sample channel. We showed the biosensing capabilities of our optofluidic YI using streptavidin as a biomolecular analyte and biotin as a recognition element. To show that biosensing is selective, we used bovine serum albumin (BSA) as an interferent and detected streptavidin in the presence of BSA. 

## 2. Experimental Section

### 2.1. Chemicals and Materials

3 mm thick clear poly (methyl methacrylate) (PMMA) sheets were purchased from RS Components (Corby, UK), and 275 µm thick double-sided tape (3M 7961MP) was obtained from Cadillac Plastics (Swindon, UK). Agarose (A9539), biotin hydrazide (B7639), sodium phosphate monobasic, sodium phosphate dibasic, fluorescein, and bovine serum albumin (BSA, A7638) were purchased from Sigma-Aldrich. Sodium (meta)periodate (NaIO_4_) and 1 N hydrochloric acid were obtained from VWR. Streptavidin (2-0203-100) was purchased from IBA Lifesciences. Lucifer yellow hydrazide (L453) was obtained from Thermo Fisher Scientific.

### 2.2. Synthesis of Functionalized Agarose

In total, 2% (w:v) agarose solution was prepared in 10 mM phosphate buffer, pH 5.3, and dissolved by placing the solution on a hotplate at 120 °C while being stirred with a magnetic bar. The agarose solution was cooled to 70 °C. To oxidize hydroxide groups in agarose to aldehydes, equal volumes of 2% (w:v) agarose solution and a 40 mM NaIO_4_ solution were reacted for 45 min at 70 °C in darkness. The solution was gelled by cooling to room temperature. The hydrogel was washed in 10 mM phosphate buffer, pH 5.3, for 4–6 h and then in 10 mM phosphate buffer, pH 6.2, overnight. The activated hydrogel was melted by heating to 70 °C and reacted with 2 mM biotin hydrazide for 2 h to obtain biotin-functionalized agarose. Alternatively, instead of biotin hydrazide, lucifer yellow hydrazide was reacted with sodium periodate-activated agarose solution. In both cases, the solution was gelled by cooling to room temperature before washing in the buffer used to prepare sample solutions. Agarose hydrogel was melted again by heating, and the solution was used to fill selected regions of channels in microfluidic devices.

### 2.3. Fabrication of Microfluidic Devices

Microfluidic devices with two designs were used; one of these devices was used to study electrokinetic transport (shown in Figure 1a), and the other was used for electrokinetic-coupled biosensing (shown in Figure 1b). Microfluidic devices were made by sandwiching a doubled-sided adhesive film between 38.1 mm long and 25.4 mm wide PMMA sheets. The top PMMA sheet had through holes to act as fluidic reservoirs. The double-sided adhesive was laser-cut to form a sensor and reference channels, each with a width and length of 1 mm and 10 mm, respectively. The center-to-center distance between the channels was 2 mm (see Figure 1 for detailed dimensions).

### 2.4. Instrumentation for Interferometry with Electrokinetic Transport

A schematic of our YI instrumentation is provided in Figure 2. The laser light of peak wavelength 532 nm (RLD 532-1-3, Roithner Lasertechnik, Vienna, Austria) was used. The power of the laser was 1 mW, and the beam diameter was 5 mm. The laser beam was passed through a beam expander (Comar Optics, Suffolk, UK) to increase the diameter of the beam to 25 mm, resulting in a power density of ~0.2 mW/cm^2^. Subsequently, the beam was passed through a Fresnel biprism with an apex angle of 179° (3B Scientific, Somerset, UK) to split it into two beams, which were passed through a cylindrical lens (Comar Optics) with a focal length of 120 mm to obtain two wedge-shaped beams separated by 2.17 mm. The distance between the Fresnel biprism and the cylindrical lens was ~325 mm. 

The microfluidic device was placed 120 mm away from the cylindrical lens. The l-shaped electrodes with two legs, which were made of 0.1 mm thick stainless steel sheets, were immersed in the reservoirs of the channels. As shown in the inset in Figure 2, electrodes were screwed on matching through holes in the microfluidic device using M4 screws and threaded nylon spacers (RS Components, Corby, UK).

Mirrors (PFSQ10-03-F01, Thorlabs, Ely, UK) were used to fold the beam to reduce the footprint of the set-up. The interferogram was captured using a 6.6 Mpixel CMOS camera (PL-B781, Pixelink, Ottawa, ON, Canada) with 10.5 mm by the 7.7 mm imaging region. The size of each pixel was 3.5 µm by 3.5 µm. The long axis of the camera was aligned parallel to the length of the channels to maximize the length of the channel to be imaged. The camera was placed ~680 mm away from the microfluidic device. Voltage was applied across the length of the channels using a power supply (PS350, Stanford Research Systems, Sunnyvale, CA, USA).

### 2.5. Instrumentation for Fluorescence Imaging

Microfluidic devices were illuminated at a ~45° angle of incidence from the bottom with a 25 mm diameter collimated light beam of wavelength 490 nm obtained using a combination of LED (Roithner Lasertechnik) and a planoconvex lens (Comar Optics). Images were captured by placing an optical bandpass filter (peak wavelength: 520 nm and bandwidth: 10 nm, Knight Optical, Kent, UK) and a camera (PL-B781, Pixelink) on the top of the microfluidic device parallel to the normal of the device surface. Fluorescence images were analyzed using the free software ImageJ 1.51.

## 3. Results and Discussion

### 3.1. Working Principle of the YI

In our YI (see Figure 2 for instrumentation set-up), light beams passed through the sensor and reference channels and then combined in space to produce a periodic pattern of intensity (*I*(*s*,*y*)) (see Figure 3) given by Equation (1) [14].
(1)$$I\left(s,\;y\right)=I_{0}+I_{0}cos[\frac{2\pi d}{\lambda D}s+\Delta \varphi (y)]$$

Here, *I*_0_ is the intensity of the incoming beam, *d* is the distance between the two beams, *λ* is the wavelength, *D* is the distance between the microfluidic device and camera, *s* is the distance along the camera and Δ*φ*(*y*) is the phase difference between the sensor and reference channels at distance y along their length and is given by Equation (2) [14] where *h* is the depth of the channels and Δ*n*(*y*) is the RI difference between the sensor and reference channels. The optical pathlength of our YI was determined by *h* and could be easily tailored.
(2)$$∆\varphi (y)=\frac{2\pi h}{\lambda }\Delta n(y)$$

As proteins are introduced at location *y* in the sensor channel, Δ*n*(*y*) changes, causing Δ*φ*(*y*) and the position of interference fringes at y to change. To obtain Δ*φ*(*y*), we extracted intensity distribution along s at the selected y and applied a fast Fourier transform (FFT) [14]. By measuring Δ*φ*(*y*), we were able to measure RI differences as low as 2.04 × 10^−6^ per mm [5]. We analyzed interference fringes along the channel length recorded at different times to obtain the RI difference profile *versus* space and time. The interference fringes were processed in real time using the algorithm reported by the authors previously [14] with a time resolution of ~4 s.

### 3.2. Electrokinetic Transport of Streptavidin Using the YI

As binding between biomolecular analytes and recognition elements can be influenced by the pH of buffer solutions [21], we studied the electrokinetic transport of streptavidin at pH values 3, 7, and 9. Microfluidic devices with sensors and reference channels filled with 1% (w:v) untreated agarose were used. The reservoirs of the reference channel and the reservoir of the sensor channel were filled with a 10 mM phosphate buffer. The other reservoir of the sensor channel was filled with 20 µM streptavidin prepared in the same buffer. When 10 V was applied across the channel length, streptavidin was electrokinetically transported because it was electrically charged at the buffer pH. Electrokinetic transport is a result of the electrostatic force experienced by a charged species in the electric field generated by applying a voltage across the length of the channels [22]. The electrostatic force is balanced by the viscous drag force, causing the charged species to move with a constant velocity in the presence of an electric field [22]. 

The phase difference *versus* channel length and time for the protein in 10 mM phosphate buffer at pH 3 is shown in Figure 4a. As time progressed, the protein traveled in the channel. Thus, the change in RI and the phase difference was propagated along the channel length. The isoelectric point of streptavidin was ~5 [23], and hence, the protein was expected to be positively charged. As the electrostatic force on a positively charged species is in the direction of the electric field, positively charged streptavidin moves toward the cathode. Thus, the phase change in the channel progressed from the anode to the cathode. 

Similar studies were performed on the protein in 10 mM phosphate buffer at pH 7 and 9. The direction of travel of streptavidin at pH 3 was reversed compared to pH 7 and 9, as it had an isoelectric point of ~5 [23]. The distance and time at which the phase difference changed to 50% of its maximum value were calculated and plotted for streptavidin prepared in buffers of pH 3, 7, and 9 (see Figure 4b). The slope of the best-fit line provided the electrokinetic velocity of the protein, which at pH 3, was 8.1 × 10^−3^ mm/s, and at pH 7, was −12.2 × 10^−3^ mm/s, and pH 9 was −15.0 × 10^−3^ mm/s.

### 3.3. Electrokinetic Transport-Coupled YI for Biosensing

We used microfluidic devices with the reference channel filled with 1% (w:v) untreated agarose, and the sensor channel contained a plug of functionalized agarose sandwiched between untreated agarose (both: 1% (w:v)). 

To determine the width of the functionalized agarose plug, fluorescein was added to the solution of functionalized agarose, injected in the sensor channel, and fluorescence images were taken (see inset in Figure 5) using the instrumentation described in Section 2.5. In Figure 5, the region with a high gray scale value corresponds to the plug of functionalized agarose containing fluorescein. Based on Figure 3, the full-width half maximum (FWHM) of the functionalized agarose plug was 2.5 ± 0.5 mm (repeats = 5).

To estimate the concentration of biotin in functionalized agarose, we used the same immobilization chemistry but replaced biotin hydrazide with lucifer yellow hydrazide. Lucifer yellow functionalized agarose solution was filled in channels of microfluidic devices, the fluorescence image was recorded, and the gray scale value in the channel was obtained using Image J. The gray scale value was converted to the concentration of lucifer yellow immobilized in functionalized agarose using a calibration curve. The calibration curve was constructed by filling microfluidic channels with lucifer yellow solutions of different concentrations, taking fluorescence images, and extracting the gray scale value in the channel for each concentration. Using this approach, the concentration of lucifer yellow immobilized in functionalized agarose was determined to be ~24 µM. Thus, we expected the concentration of biotin in functionalized agarose to be ~24 µM.

Subsequently, a new microfluidic device was prepared where the sensor channel was filled with biotin-functionalized agarose sandwiched between unfunctionalized agarose. In this case, fluorescein was not used. The reference channel was filled with unfunctionalized agarose. In total, 20 µM of streptavidin prepared in a 10 mM phosphate buffer at pH 3 was electrokinetically transported in the sensor channel of the prepared microfluidic device. Figure 6a shows that, initially, streptavidin traveled from one end of the channel to the other. In this initial phase, instead of traveling through the hydrogel, streptavidin might have leaked through the gaps in the sensor channel. These gaps are introduced because the hydrogel was not chemically bonded to the top and bottom PMMA surfaces of the channel and, hence, could detach as it solidified. Subsequently, streptavidin flowed through the hydrogel in the channel and accumulated in the region of the channel containing the biotin-functionalized agarose plug. 

After a buffer wash was conducted, streptavidin stayed bound to the biotin-functionalized agarose plug, as illustrated in Figure 6. This is because the interactions between streptavidin and biotin are some of the strongest, with a dissociation constant of ~10 fM. The phase difference in regions upstream and downstream of the biotin-functionalized plug (black and blue traces in Figure 6b) after the buffer wash returned to the value at time = 0, indicating that streptavidin did not bind to untreated agarose. The average phase difference between 2500 s to the end was plotted with respect to the distance to determine the FWHM of the plug of bound streptavidin. The FWHM of the bound streptavidin plug was ~3.1 mm, which was comparable to the FWHM of the agarose plug determined by fluorescence images.

The sensing of streptavidin at pH 7 and 9 were also studied. However, the phase difference after buffer wash in the region corresponding to biotin-functionalized agarose was low (<0.4 *versus* 1 radian at pH 3). This was unexpected because biotin and streptavidin binding was strong over a wide pH range. 

### 3.4. Effect of Interferents on Sensing of Streptavidin

Figure 7 shows streptavidin sensing in the presence of BSA. Figure 7a shows that the phase difference increased as BSA was electrokinetically transported in the sensor channel but returned to the values observed at time = 0 after a buffer wash. Thus, BSA did not bind to functionalized biotin and untreated agarose. However, as shown in Figure 7b, if a mixture of BSA and streptavidin was electrokinetically transported, there was a permanent increase in the phase difference in the region of the sensor channel with a biotin-functionalized agarose plug.

### 3.5. Quantification of Streptavidin

Figure 8 shows that 2 µM of streptavidin in a 10 mM phosphate buffer at pH 3 accumulated in the biotin-functionalized agarose plug. Streptavidin could not travel beyond the biotin-functionalized agarose plug. A comparison of Figs. 6 and 8 shows that the phase change in the biotin-functionalized agarose plug after the buffer wash was ~1 radians for both 20 and 2 µM streptavidin. This is because the concentration of biotin in the plug was ~24 µM. Hence, at the selected concentrations of streptavidin (i.e., 2 and 20 µM), all the immobilized biotins bound to streptavidin because of the low dissociation constant (K_D_) of ~10 fM.

The RI LOD of our optofluidic YI was 2.04 × 10^−6^ RIU per mm of the optical pathlength [14]. In this work, we used 275 µm deep channels, and the protein RI increment was 0.185 mL/g [24]. Thus, the LOD of our YI for streptavidin was ~0.6 µM in the absence of biotin-functionalized agarose in microfluidic channels. In the presence of biotin-functionalized agarose, in theory, our optofluidic YI should have been able to detect 2.5 fM of streptavidin. However, experimentally, this could not be achieved. The dynamic range was limited by the K_D_ between the analyte and recognition element, though not by the YI, as phase unwrapping allows the detection of unlimited changes in RI, provided that they do not happen suddenly.

### 3.6. Comparison with State-of-the-Art Interferometers

Table 1 highlights that the RI LOD of our optofluidic YI is comparable to waveguide MZI. The RI LOD of our optofluidic YI is about 30 and 800 times higher than the bimodal waveguide and waveguide YI, respectively. However, based on Equation (2), the phase change is linearly proportional to the optical pathlength. This implies that RI LOD can be improved by increasing the pathlength, which can be easily achieved for our optofluidic YI using deeper microfluidic channels. Thus, a prudent way of comparing different interferometers would be to compare their RI LOD per unit optical pathlength. Based on the RI LOD per unit optical pathlength, our optofluidic YI is comparable to bimodal waveguide and is 10 times better than waveguide MZI but about 50 times worse than waveguide YI. 

The current analysis time for our optofluidic YI is comparable to the state-of-the-art devices but can be significantly reduced by increasing the voltage used for electrokinetic transport. All results presented in this work were obtained by applying 10 V across the channels of microfluidic devices. The analysis time of our optofluidic YI could easily be reduced to about 10 min using about 20 V. 

Furthermore, the RI imaging capability of our YI implies that different recognition elements can be immobilized along the length of channels for multiplexed sensing without having to make arrays of the devices. Alternatively, our optofluidic YI can be used as a whole-column RI imaging detector for monitoring electrokinetic separation methods.

Finally, our YI was made of PMMA and double-sided tape and by laser-cutting rather than expensive materials and semiconductor fabrication methods. Hence, the cost per device of our optofluidic YI was lower than state-of-the-art interferometers, which are typically made of silicon using semiconductor fabrication methods in cleanrooms.

## 4. Conclusions

An optofluidic Young interferometer (YI) coupled with electrokinetic transport was applied for the biosensing of an exemplar biomolecule, streptavidin, in the absence and presence of bovine serum albumin interferent. While electrokinetic forces allowed the rapid transport of samples, YI provided temporal and spatial information on the concentration and interactions of biomolecules with recognition elements. Our optofluidic YI provides a one-dimensional refractive index image, which contrasts with current interferometers that are point detectors and, hence, is highly suited for multiplexed detection with recognition elements immobilized at different locations along the length of microfluidic channels. Furthermore, the optical pathlength and, hence, refractive index limit of detection for our optofluidic YI can be easily tailored by changing the depth of microfluidic channels rather than having to re-design waveguides, as is the case for the current interferometers. Finally, the speed of analysis could be shortened significantly by applying higher voltages to speed up electrokinetic transport. Future work will focus on applying our optofluidic YI coupled with electrokinetic transport for the sensing of the antibody–antigen and complementary DNA strand systems to ascertain the experimental limit of detection.

## Figures and Tables

**Figure 1 micromachines-15-00861-f001:**
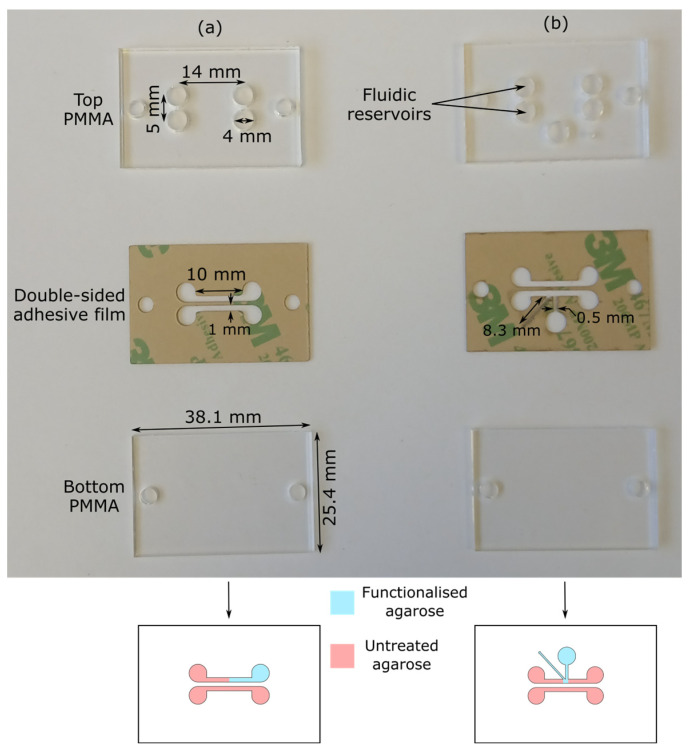
Images and detailed dimensions of the top PMMA, double-sided adhesive, and bottom PMMA used to make microfluidic devices to study (**a**) the electrokinetic transport and (**b**) electrokinetic transport-coupled biosensing.

**Figure 2 micromachines-15-00861-f002:**
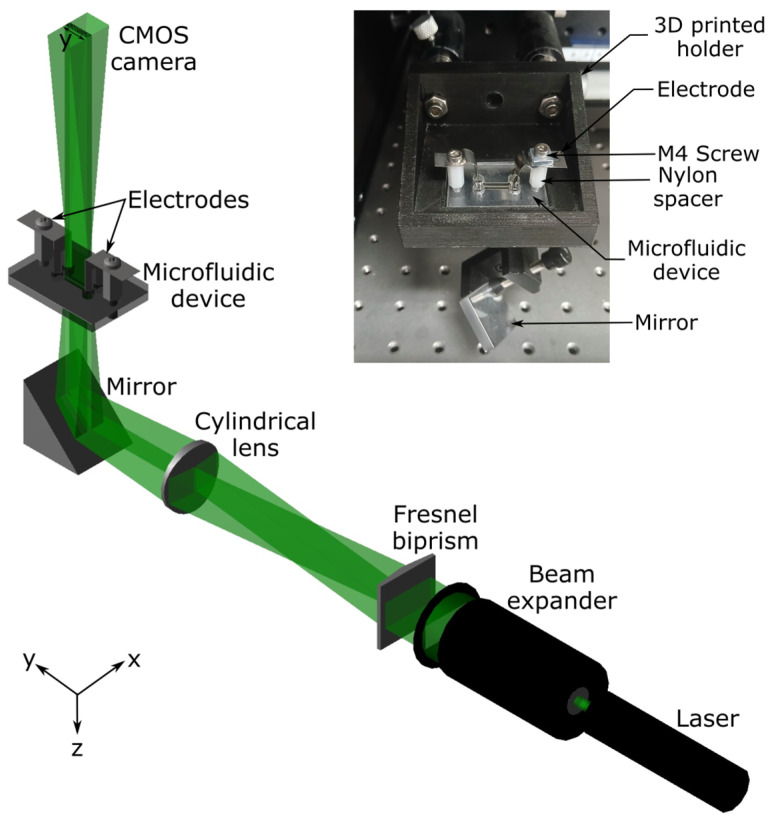
Schematic of the YI instrumentation, where the inset shows microfluidic device and electrode assembly.

**Figure 3 micromachines-15-00861-f003:**
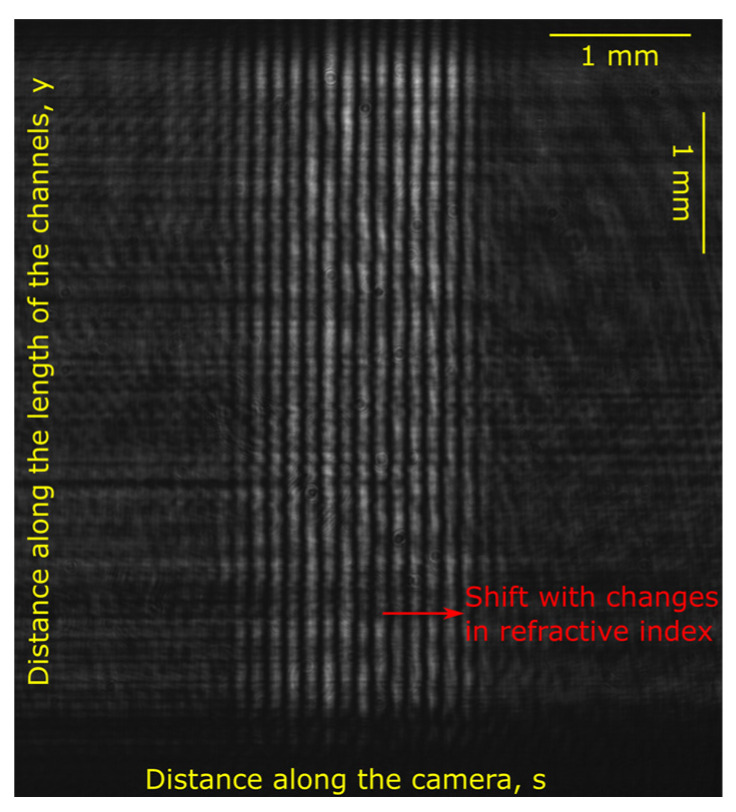
Typical interference fringes for our optofluidic YI.

**Figure 4 micromachines-15-00861-f004:**
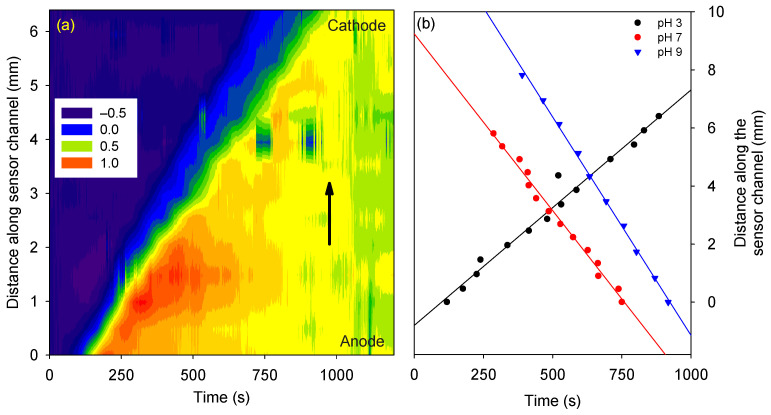
(**a**) Phase difference *versus* channel length and time (arrow shows the direction of travel for the protein prepared in buffer at pH 3), and (**b**) velocity plots of streptavidin in buffers of pH 3, 7, and 9 under 10 V.

**Figure 5 micromachines-15-00861-f005:**
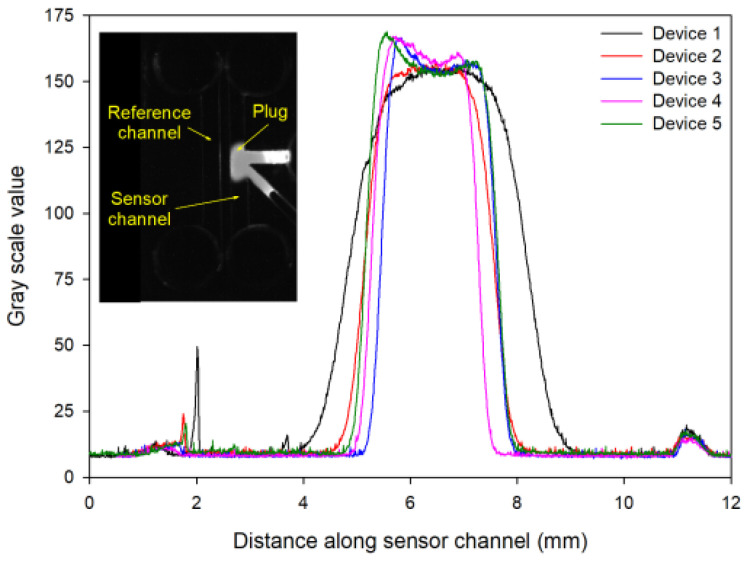
Gray scale value along the channel length for 5 microfluidic devices where the inset shows a fluorescence image of one of the devices with a 1 mm wide sensor channel filled with a plug of functionalized agarose containing fluorescein.

**Figure 6 micromachines-15-00861-f006:**
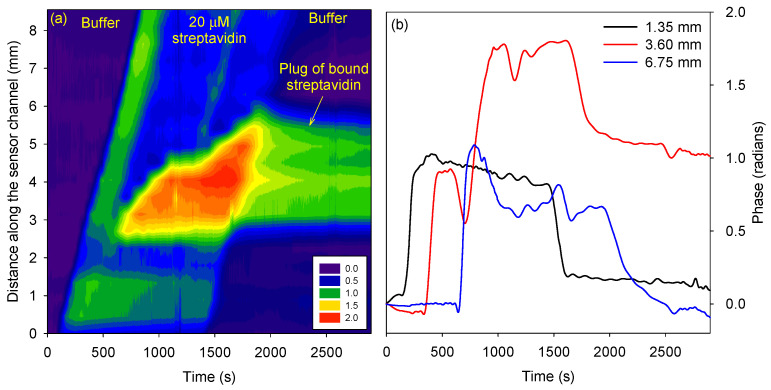
(**a**) Phase difference *versus* channel length and time, and (**b**) phase *versus* time at selected distances for streptavidin buffer.

**Figure 7 micromachines-15-00861-f007:**
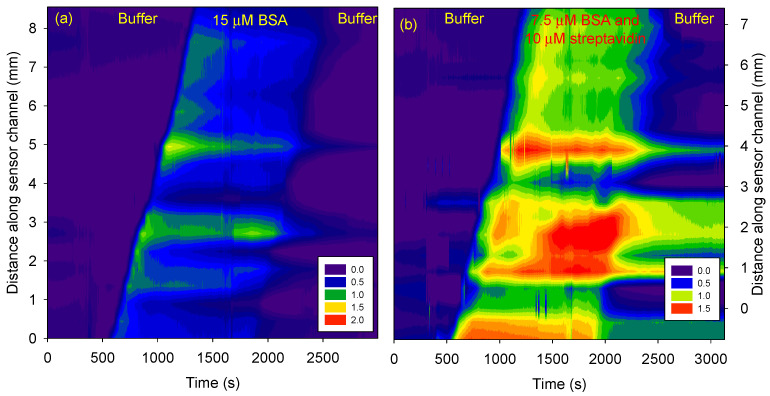
Phase difference *versus* channel length and time for (**a**) BSA buffer, and (**b**) BSA and streptavidin buffer.

**Figure 8 micromachines-15-00861-f008:**
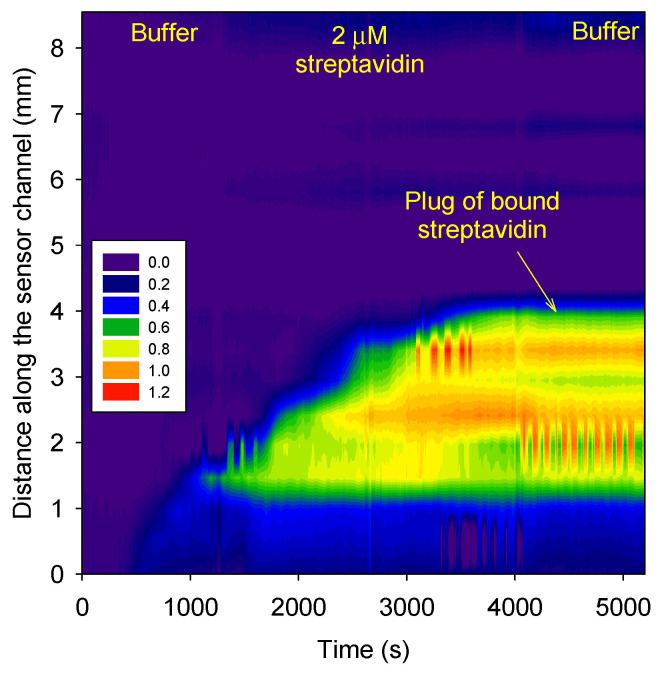
Phase difference *versus* channel length and time.

**Table 1 micromachines-15-00861-t001:** Comparison of our YI with state-of-the-art interferometers.

Type	RI LOD	Pathlength	Time (min)	RI Image	Cost per Device
Waveguide YI [11]	9 × 10^−9^	5 mm	20–50	No	High
Waveguide MZI [12]	3 × 10^−6^	10 mm	40		
Bimodal waveguide [13]	2.5 × 10^−7^	15 mm	20–30		
This work	7.5 × 10^−6^	275 µm	25	Yes	Low

## Data Availability

The original contributions presented in the study are included in the article; further inquiries can be directed to the corresponding author.

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
