# Peer review of "An Optofluidic Young Interferometer for Electrokinetic Transport-Coupled Biosensing"

_micromachines, 2024, doi:10.3390/mi15070861_

Round 1
Reviewer 1 Report
Comments and Suggestions for Authors
The paper presents a label free interferometric method for label-free biosensing. A Young's interferometer setup constructed with a plane wave excitation of a Fresnel bi-prism is used for this purpose. The approach is innovative the the problem the paper addresses is very relevant to needs of biotechnology. The work should be of interest to the readers of the journal. However, there are a few things that should be addressed before the paper can be considered for publication. My detailed comments are listed below:
1. A schematic of the device geometry is missing. Please include a schematic that shows the full structure of the device, including the fluid channels and the electrodes. The dimensions should be labeled. The schematic can be accompanied by a photograph of the device.
2. The approach of the paper is to reconstruct phase information from the intensity value of a grayscale image that is captured by the camera during the experiment. However, only the processed phase data is shown in the figure. I think both the raw data (the image) and the processed phase data should be shown side by side. That will improve the presentation of the paper.
3. The results are mostly presented in a time-space plot with the color indicating the phase difference (Fig. 2a, 4a, 5 and 6). These are fine. However, a single snapshot of time with both x and y axis being spatial dimension (space-space plot with color indicating phase difference) would be a very interesting result. This will be a direct one-to-one analog of the raw grayscale image, and showing them side by side should be considered.
4. What is the time resolution that can be achieved?
5. How fast is the data processing algorithm/code? Would it be possible to do the processing in real time? As biosensing cases are usually time dependent and can sometimes changes can occur in very small time scales, real-time sensing can be of extreme importance. The authors may choose to focus on this in their future work.
6. In section 3.3, the manuscript discusses fluorescence images. Fluorescence signals can be large and one would assume that they would interfere with the fringe pattern and thus introduce errors in phase retrieval. Please discuss what actions were taken to avoid this. Perhaps limiting the fluorescence signal (with low concentration) or some post-processing algorithm?
7. It is mentioned in page 5 "These gaps are introduced because the hydrogel was not chemically bonded to the top and bottom PMMA surfaces of the channel and hence can detach as it solidifies". Was this intended? If not, can this be avoided?
8. Could the authors please discuss further about why the biotin streptavidin binding was found to be low in the pH 7 to 9 range? Was this verified by a different benchtop measurement?
9. It is not very clear how he stainless-steel sheet electrodes were fixed to the device surface. Could the authors elaborate?
10. Do the physical structures of the device interfere with the fringes and cause issues in the phase retrieval?
11. Please indicate the power of the laser.
12. The electrokinetic transport phenomenon is not sufficiently discussed. A bit more discussion on the topic would help improve the readability of the paper.
13. I do not see any illumination light in the system besides the laser. Although ideal for quantitative measurements, laser illumination is usually not ideal for taking optical image of the device surface. Usually, systems employ a separate white light illumination for imaging purposes which is turned off during the measurement period. Please discuss if such a light was used or if a different technique was used for optical inspection.
14. The reference section of the manuscript is inadequate. Since the work discusses optical biosensing, some other optical biosensing techniques should be briefly mentioned and cited in the introduction section. For example, plasmonic sensing and spectroscopy based methods. I recommend the following:
a. https://doi.org/10.1021/acs.chemrev.8b00359
b. https://doi.org/10.1063/5.0191871
c. https://doi.org/10.1002/adma.201200373
English is fine.
Reviewer 2 Report
Comments and Suggestions for Authors
The authors reported a novel optofluidic Young interferometer (YI) that provides real-time spatial information on refractive index changes occurring along the length of sensor and reference channels for streptavidin detection. However, some issues must be cleared
-It is necessary to establish the biotin immobilization method in detail.
-Some tests were performed to estimate the amount immobilized.
-Has any testing been done to determine if biotin is lost during washing?
-The authors must develop a more in-depth discussion of the possible causes that affected the device's detection limit so abruptly compared to the theoretical one (~0.6 µM with respect to 2.5 fM).
-Furthermore, it is necessary to establish how the detection limit was calculated experimentally. During the manuscript, streptavidin concentrations of 10, 20, and 2 µM were measured. Were more concentrations measured to make a calibration curve?
-Additionally, it is necessary to explain in detail why there was no difference in the measurement of 20 and 2 µM (lines 204 to 206) because this impacts the device's low sensitivity.
-In section 3.5, "Comparison with latest generation interferometers," it is necessary to carry out a more in-depth discussion of the analytical parameters reported in Table 1. What is the importance of the Pathlength? On what basis is it established whether the cost of the device is low or high? What is the difference between the device designs? Are there any settings that favor better detection limits?
+The conclusion must be improved by considering the device's advantages and disadvantages in relation to what is reported in the literature, as well as the detection limit obtained and its difference from what is theoretically established.
Round 2
Reviewer 1 Report
Comments and Suggestions for Authors
The paper has been improved. Although the authors did not incorporate all of the suggested modifications, it is, however, still satisfactory. It can be considered for publication now.
Comments on the Quality of English LanguageEnglish is fine
Reviewer 2 Report
Comments and Suggestions for Authors
The authors have fully responded to the issues and comments made during the first review round. Therefore, it is suggested that the study be accepted.